# The Role of the Nurse in the Care and Management of Patients with Rheumatic Diseases Arising from the Current EULAR Recommendations: A Literature Review

**DOI:** 10.3390/healthcare11172434

**Published:** 2023-08-31

**Authors:** Anna Bednarek, Robert Klepacz, Iwona Elżbieta Bodys-Cupak

**Affiliations:** 1Department of Health Promotion, Faculty of Health Sciences, Medical University of Lublin, 20-093 Lublin, Poland; 2Department of Clinical Pathomorphology, Medical University of Lublin, 20-093 Lublin, Poland; robert.klepacz@umlub.pl; 3Department of Nursing Fundamentals, Institute of Nursing and Midwifery, Faculty of Health Sciences, Jagiellonian University Medical College, 31-501 Krakow, Poland; i.bodys-cupak@uj.edu.pl

**Keywords:** rheumatic diseases, nursing care, advanced practice nurse, EULAR recommendations for nurses

## Abstract

In some countries, restructuring of the healthcare system has contributed to the introduction of a new domain in professional nursing—the Advanced Practice Nurse (APN). In order to provide the highest quality of care to patients, nurses working at the advanced level are expected to develop knowledge and to initiate services and practices carried out in collaboration with other professionals. In 2018, the European League Against Rheumatism (EULAR) updated its recommendations for the role of the nurse in the management and care of patients with rheumatic conditions. The objective of the study was a presentation of the scope of medical services provided by nurses for patients with rheumatic diseases based on current EULAR recommendations. A review of the literature on the participation of nurses, as members of a multidisciplinary team, in the education, management, psychosocial support, and promotion of self-care in patients with rheumatic diseases was presented. The expert group formulated three overarching principles and eight recommendations. The literature review and expert recommendations indicated that nurses’ tasks in relation to patients with rheumatic diseases should include an initial assessment of health needs, routine follow-up care, and counseling for patients on self-care and lifestyle changes. In the EULAR recommendations, nursing care is also defined as a practice model in which nurses, in collaboration with physicians, provide support, education, and disease monitoring to patients with rheumatic conditions. The requirement for extended nursing education at the advanced practice level, aimed at acquiring diagnostic, therapeutic, caring, and educational knowledge and skills was highlighted, particularly with regard to the EULAR recommendations.

## 1. Introduction

The global prevalence of rheumatic diseases in adults ranges from 0.24% to 1% and is varied by genetic and environmental factors. Rheumatic diseases are also a significant cause of chronic illnesses in the pediatric population. Multiple organ complications, disability, and the inability to participate in social life are common consequences [1,2]. Public education programs, introduced in many countries, seek to increase public awareness of rheumatic diseases [3,4]. They have also contributed to early diagnosis and therapy. On the other hand, advances in the management of rheumatologic conditions and biomedical research in immunology and genetics in recent years have raised the prospect of optimal health functioning for patients affected by inflammatory joint disease.

Most people want to maintain a high quality of life despite chronic illness. This has resulted in a shift in the health needs of patients and their families, particularly in the need for professional support and preparation for self-care. Hence, new models of care, including innovative organizational and educational solutions for nurses, have also become necessary [5].

The need for long-term multispecialty care of chronically ill patients, including those with rheumatic conditions, has resulted in the development of a new role for nurses, i.e., the Advanced Practice Nurse (APN), with in-depth knowledge in a specialized area of medicine and in social sciences, with expanded clinical skills in diagnosing, treating, and following up (co-managing) patients. The professional role of an APN was first described in the United States in the 1970s. The International Council of Nurses defined an APN as a nurse with the expertise and ability to make the independent decisions necessary for extended practice. APNs currently integrate research, education, practice, and care management. They have a high level of professional independence and advanced patient health assessment skills. In addition, APNs independently plan, implement, and evaluate health programs and provide consultation services to other entities in the healthcare system [6].

Internationally, there is ample evidence of the positive impact APNs have on patients, healthcare organizations, and the nursing profession. In many countries, there are pilot programs which aim to introduce the new role of the APN into national healthcare systems. These programs are constantly developing in order to incorporate even more clinical skills and practical experience into nursing studies. It is also important to introduce appropriate regulations to allow APNs to function as part of a multidisciplinary healthcare team along with other stakeholders involved in the care of chronically ill patients and their families [7,8,9].

The need for, and scope of, the interdisciplinary care provided by nurses is determined by the health problems of patients with rheumatic diseases. The implementation of EULAR recommendations into the practices carried out by nurses for patients with rheumatic diseases is based on the specialized education of rheumatology nurses [10].

The management of patients with rheumatic diseases has changed dramatically over the last decade and involves the early initiation of intensive treatment and the close monitoring of disease activity until remission is achieved. Similarly, in less common connective tissue diseases, which are characterized by immune dysfunction, vascular damage, and changes to many internal organs, the treatment of patients requires a comprehensive approach.

Due to the large spectrum of clinical symptoms that affect the quality of life of patients with rheumatic diseases, including connective tissue disease, professional nursing care is essential. It provides an individual approach to patients and gives them a sense of security and confidence in the disease. In view of the above data, the improvement of services and areas of competence for rheumatology nursing has become necessary [11].

Recognizing the new roles of the nurse in relation to patients with rheumatologic conditions, the European League Against Rheumatism (EULAR), whose overarching goal is to improve the treatment and care of people with musculoskeletal conditions in European countries, in 2018 re-evaluated its recommendations regarding the nurse’s role in the treatment and care of patients with chronic arthritis using current scientific evidence.

The aim of this paper is to present the scope of medical services provided by nurses for patients with rheumatic diseases based on the current EULAR recommendations.

## 2. Methods Review

### 2.1. Study Design

Since a systematic review according to PRISMA reporting standards did not show the expected results, a literature review was adopted as the study design [12]. The chosen review style was a literature review, given the uncertain nature of any eligible studies.

### 2.2. Search Methods

Standard online search tools (https://www.equator-network.org/; accessed on 15 May 2023) were used to compile the review. PRISMA and AGREE did not yield results. This indicates a lack of studies covering the tasks of the nurse in the implementation of care for patients with rheumatic diseases from current (after 2018) EULAR recommendations. Therefore, the PRISMA 2023 flow chart for new systematic reviews, which includes searching databases, registries, and other sources, was not performed. All authors reviewed the databases and obtained the same results. Further considerations were based on the literature review obtained, according to the accepted keywords.

The article was based on the literature review, including the recommendations of the EULAR Panel of Experts, on the role of the nurse in the education, psychosocial support, and management of rheumatological patients.

In 2012, the first EULAR recommendations for rheumatology nurses were published, raising the importance of their participation in the care, treatment, monitoring, and diagnosis of patients with rheumatoid diseases, including connective tissue diseases. These recommendations were modified and supplemented in 2018 on the basis of research that has appeared since the last recommendations were issued and based on the experience of people making up the team of doctors caring for a rheumatological patient, as well as the patients struggling with rheumatological disease themselves. Therefore, the search for papers in scientific resource databases started from the period of the first EULAR recommendations in 2012.

In May 2023, we searched the PubMed database. The search covered scientific medical publications from January 2012 to 15 May 2023 and the following keywords were used: “rheumatic diseases”, “rheumatology nursing”, “advanced nursing practice”, “EULAR recommendations for nurses”. The search for articles for analysis did not include studies from before 2012, articles written in languages other than English, studies on the management of rheumatological patients not related to the participation of nurses, or post-conference abstracts not related to the participation of nurses.

Figure 1 outlines our process of selecting articles for review according to the adopted keywords. At each stage, an article was selected if it was identified for inclusion by all authors, based on the eligibility and exclusion criteria. Ultimately, 47 full-text articles were identified for analysis and synthesized.


*Inclusion criteria*


The inclusion criteria were peer-reviewed articles published in English, in the period from January 2012 to 15 May 2023, which discussed the topic of advanced nursing care for patients with rheumatological diseases, based on EULAR recommendations.


*Exclusion criteria*


Articles were excluded if the research was: (1) outside the scope of nursing care; (2) non-empirical; or (3) not published in English.

### 2.3. Search Outcome for EULAR Recommendations

EULAR’s recommendations were prepared using an updated version of the EULAR standardized operating procedures for the development, evaluation, dissemination, and implementation of recommendations [13,14]. A five-member steering committee set up a team of experts representing 17 European countries (task force), including 15 nurses, 2 patients, a physiotherapist, a psychologist, an occupational therapist, a medical student, and 3 rheumatologists, one of whom was also a methodologist.

The strategy of the literature review assumed that the search would cover the period from August 2012 to 1 December 2020. The search identified 2609 records, out of which 51 studies were selected (41 articles and 10 abstracts), mostly related to rheumatoid arthritis. The selected items included 2 meta-analyses, randomized controlled trials (14 articles), observational studies (20 articles), 8 pieces of survey research, 2 mixed-method studies, and 5 qualitative studies. The following selection criteria were used: only the studies that provided evidence at a higher level than the first recommendations and included references to rheumatology nursing were included. The steering committee performed a critical assessment of the quality of the articles using the following measurement tools: *AMSTAR II* for literature reviews, *Cochrane Risk of Bias* for randomized controlled trials, *Quality In Prognosis Studies (QUIPS)* for observational studies, and, in the analysis that involved economic evaluations, *Consensus on Health Economic Criteria* [14,15]. The evidence was divided into four grades of recommendations, from A (the highest) to D (the lowest), and five levels of evidence (1–5; high–low) according to the Oxford Center for Evidence-Based Medicine [16]. The level of agreement for each updated recommendation was rated using a numerical rating scale from 0 to 10 (0 = “do not agree at all” and 10 = “fully agree”). The resulting level of agreement among task force members was high (mean 9.7; range 9.6–10.0).

We included all studies related to the nursing care of patients with rheumatologic diseases, and, in particular, the tasks defined in the latest EULAR recommendations. We did not impose any geographic restrictions; we only included papers published in English in the years designated by subsequent EULAR recommendations.

Using the same tools, criteria, and keywords, the authors performed independent database searches. This study focuses on recommendations that aim to shape the way nursing care is to be delivered in the future. The practical significance of the EULAR recommendations will become apparent several years after their implementation, and after empirical studies present both the subjective and objective scope of activities performed on the basis of the presented guidelines.

We classified the content following the order of the EULAR recommendations. We attempted to present the changes introduced by the new 2018 recommendations. We summarized the recommendations in a table, including the scoring. One author reviewed each paper, which was then checked and discussed with a second reviewer. We revised the recommendations on the basis of the literature review (review, comments) to present the roles of the nurse in providing care to the rheumatology patient, emphasizing the specific tasks dedicated and assigned to the Advanced Practice Nurse.

## 3. Results

The first EULAR recommendations on the role of the nurse in the management of patients with rheumatic diseases, including rheumatoid arthritis, ankylosing spondylitis, psoriatic arthritis, and spondyloarthropathy, were published in 2012 [17]. The ten recommendations formulated at that time defined the professional nursing tasks in the care of the patient with rheumatological disorders to be applied in all European countries [18]. An evaluation of these recommendations revealed a high level of agreement among the expert group members from different countries, but it also revealed significant differences in their application, suggesting that these recommendations are not widely known and implemented. Moreover, some recommendations were based on a low level of evidence [18,19,20].

In 2018, the EULAR expert group reached a consensus on the final wording of the updated recommendations and formulated a total of three overarching principles and eight recommendations (Table 1).

The three overarching principles, as general guidelines, apply to all recommendations and emphasize the nurse’s role as an integral member of the healthcare team who provides evidence-based care and supports shared decision-making when consulting a patient. The eight recommendations relate to the participation of the rheumatology nurse in needs-based patient education, and in ensuring satisfaction with care and prompt access to various services. In addition, an important task for the nurse is to monitor the disease and the effectiveness of care. Psychosocial support for the patients and their families is also important, as is encouraging them to promote self-care (disease self-management). The EULAR 2018 recommendations on the role of the nurse in the management of rheumatology patients have been updated according to the available evidence. The current update represents a stronger consensus among experts than the previous recommendations from 2012.

## 4. Discussion

### 4.1. EULAR Recommendations for Nurses, Taking into Account the Literature Review

A review of the literature indicates the effectiveness, safety, and accessibility of care provided by nurses from the patient’s perspective [9,10]; this is also reflected in the priority principles for the recommendations developed by the EULAR 2018 panel of experts. Rheumatology nurses work in close collaboration with patients and their families [17]. As members of an interdisciplinary healthcare team, they share a focus on effective care that addresses health needs, values, and patient preferences. In their professional practice, nurses use different sources of knowledge, i.e., scientific evidence, including protocols and guidelines, but also their own nursing experience, patients’ aspirations (needs), and territorial contexts [21,22].

Patient education by the nurse, including therapeutic aspects and health promotion, is strongly emphasized in the updated EULAR 2018 recommendations [23]. Several authors confirm the positive impact of nurse-led patient education programs in disease-related areas such as pain, illness perception, quality of life, and adherence to recommendations [22,23,24,25].

Guided by evidence from the research, patient satisfaction with nursing care was made a priority. It was demonstrated that consultations with nurses have a significant, positive, and long-term (two years) effect on patient satisfaction [26]. In addition, it was reported that patients appreciate maintaining a professional relationship with a nurse. Studies also emphasized nurses’ holistic and professional approach to care, and patients’ trust in nurses’ knowledge, as well as a supportive style of mutual relations [27,28,29].

The unpredictable nature of rheumatic conditions and new treatment options sometimes require prompt access to care. At the same time, we can take advantage of the new forms of communication, support, and follow-up of diseases that are provided by online consultations, which allow for remote care [28]. When making their recommendation for quick patient access to nursing care, the EULAR task force focused on the current scientific evidence from qualitative studies in which patients indicated that telehealth follow-up and online consultations by nurses gave them a sense of personalized support from a knowledgeable healthcare team. The quality of this kind of service was comparable to the quality of a conventional follow-up to control the disease activity performed by a physician [30,31].

Randomized clinical trials comparing nurse-led and physician-led follow-up of patients showed that the quality of control of disease activity resulting from nursing care was equivalent to the quality of control when the care was provided by physicians. There were no significant differences in the quality of health functioning between patients monitored by a nurse and a physician. In addition, nurses played an important role in the early detection of inflammatory changes in joints (by physical examination, including assessment of pain, swelling and skin warmth within the joint), interpretation of laboratory results, continuation of drug treatment, and adherence to vaccination schedules [32]. Nursing care was cost-neutral or slightly less costly than the care provided by physicians, but there was no evidence of savings in services provided. When developing this recommendation, the panel of experts recognized the need for involving nurses in the comprehensive management and monitoring of the disease (disease management) [33,34,35].

Further recommendations of the expert panel focused on the need for nurses to provide support to patients with psychosocial problems in order to reduce symptoms of anxiety and depression and to motivate them to function effectively despite their disease. Psychological stress experienced by patients with rheumatologic conditions has a documented negative impact on their somatic complaints and should be eliminated. In addition, depression is a well-known comorbid condition that requires proper treatment and careful handling of the patient [36].

In a study of rheumatology patients with symptoms of depression, the quality of nurse-led care was found to be equivalent to the care provided by a rheumatologist. It was noted that patients and their families appreciated the opportunity to have an in-depth conversation with a nurse, where they could tackle various issues affecting their mental state [24]. Hence, the identification of the psychosocial problems of patients with rheumatologic conditions and supporting them in this respect were found to be key components of nursing care. The simultaneous promotion of self-management gives patients and their families the opportunity to gain the necessary knowledge, skills, and confidence to cope with the physical and psychosocial consequences of living with a chronic disease and facilitates making lifestyle changes following their preferences [29]. Research has confirmed that nurse-led interventions can improve patients’ and/or families’ sense of self-efficacy to cope with fluctuations of the disease in daily life [37,38,39].

The final EULAR 2018 recommendations address the aspects of continuing education for nurses in the specialty of rheumatology in order to improve and maintain knowledge and skills, as well as the undertaking of extended/advanced roles following specialized training, in compliance with national regulations. The wording of these recommendations remained unchanged from previous recommendations [40]. However, the level of evidence for these recommendations has increased due to new insights from studies showing that educational programs for rheumatology nurses resulted in increased knowledge and skills, as well as improvements in practices. Some tasks traditionally performed by physicians and physiotherapists, such as joint examination, identification of signs and symptoms in rheumatic diseases, and the ability to distinguish abnormalities, can be performed by nurses who have completed specialized training. Rheumatology nursing is not, however, a formal specialty in every country. Yet the education of nurses in the specialty of rheumatology is crucial to the development of competent and holistic care for patients of all ages with rheumatological problems [41,42].

### 4.2. Summary of Recommendations—Tasks for Nurses in the Care of Patients with Rheumatic Diseases

The EULAR 2018 recommendations for rheumatology nurses and the review of the research highlight the need to improve nurses’ knowledge and practice of specialized clinical skills, pedagogical and psychological education, and work organization. Nurses, especially APNs, play an important role in the process of patient diagnosis and therapy (Table 2). 

At the same time, the importance of their communication skills in identifying patients’ psychosocial problems and in supporting patients and their families in making therapeutic decisions is also increasingly recognized. Person-centered care and partnership with patients are also important areas in APNs’ activities that motivate patients to self-care. The proper organization of care by nurses is important to avoid the risk of excessive workload that may reduce the quality of services provided [43,44].

**Table 2 healthcare-11-02434-t002:** Summary of selected items of the literature review in terms of the overriding principles and detailed EULAR recommendations regarding the care of rheumatological patients by nurses.

Author	Title	Journal, Year	Summary of Research Results
Perkin K. [9]	Nurse Practitioners and Interprofessional Collaboration	*Journal of Interprofessional Care* 2011, 25(4), 243–4.	In the opinion of the patient, care provided by nurses is effective, safe and easily accessible. Rheumatology nurses are important members of the medical team caring for the patient.
Riley L., et al. [11]	The Role of Nurse Practitioners in Delivering Rheumatology Care and Services: Results of a U.S. survey	*Journal of the American Association of Nurse Practitioners* 2017, 29(11),673–681.	Rheumatology nurses work in an interdisciplinary medical team, they constantly cooperate with the patient and his family in the field of care and treatment.
van Eijk-Hustings Y,. et al. [18]	Dissemination and Evaluation of the European League against Rheumatism Recommendations for the Role of the Nurse in the Management of Chronic Inflammatory Arthritis: Results of a Multinational Survey among Nurses, Rheumatologists and Patients	*Rheumatology* 2014, 53 (8), 1491–6.	In professional practice, nurses use various sources of knowledge, i.e., scientific evidence, including protocols and guidelines, as well as their own experience gained while working with patients and their families.
Fusama M., et al. [20]	Survey on Attitudes Regarding EULAR Recommendations for the Role of Nurses Involved in Medical Care of Patients with Chronic Inflammatory Arthritis in Japan	*Modern Rheumatology* 2017, 27(5), 886–93.	In patient care, nurses rely on a variety of sources of knowledge, including scientific evidence, protocols, and guidelines. They are also guided by their own nursing experience resulting from the territorial context and take into account the needs and aspirations of patients.
Barbosa L., et al. [21]	Applicability of the EULAR Recommendations on theRrole of the Nurse in the Management of Chronic Inflammatory Arthritis in Portugal	*Acta Reumatologica Portuguesa* 2013, 38,(3), 186–91.	
Solomon D.H., et al. [26]	Roles of Nurse Practitioners and Physician Assistants in Rheumatology Practices in the US	*Arthritis Care & Research* 2014, 66(7), 1108–13.	The care provided by rheumatology nurses is effective and takes into account the health needs, values and preferences of patients. Patient education by nurses covers areas related to the disease, such as pain, quality of life, and adherence.
Walker J.[44]	Rheumatoid Arthritis: Role of the Nurse and Multidisciplinary Team	*British Journal of Nursing* 2012, 21(6), 334, 336–9.	Patients are satisfied with the nursing care. They have a positive experience of maintaining a competent relationship with the nurse.
Morgan S. & Yoder L.H. [45]	A Concept Analysis of Person-Centered Care	*Journal of Holistic Nursing* 2012, 30(1), 6–15.	The holistic and professional approach of nurses to care inspires the trust of rheumatology patients in the knowledge of nurses, and also strengthens the supportive style of mutual relations.
Larsson I. [46]	Nurse-Led Care and Patients as Partners Are Essential Aspects of the Future of Rheumatology Care	*The Journal of Rheumatology* 2017, 44(6), 720–722.	In the patients’ opinion, teleconsultation by nurses provides them with a sense of individual support from a competent healthcare team. The quality of service provided by nurses is comparable to being observed by a physician in terms of disease control.
Frølund J.C. & Primdahl J. [38]	Patients’ Experiences of Nurse-Led Screening for Cardiovascular Risk in Rheumatoid Arthritis	*Musculoskeletal Care* 2015, 13(4), 236–47.	Rheumatology nurses have an important role in the early detection of inflammatory changes in the joints, the interpretation of laboratory test results, the follow-up of pharmacological treatment, and the monitoring of patients’ adherence to vaccination schedules.
Grønning K., et al. [28]	Patients’ Confidence in Coping with Arthritis after Nurse-Led Education; a Qualitative Study	*BMC Nursing* 2016, 15(28), 1–8.	Patients appreciate the opportunity to discuss various issues affecting their mental state with the nurse. Nurses provide support to patients with psychosocial problems in order to reduce the symptoms of anxiety and depression and motivate them to function effectively in the disease.
Robinson S., et al. [42]	A National Survey of Nurse Training: Confidence and Competence in Educating Patients Commencing Methotrexate Therapy	*Musculoskeletal Care* 2017, 15(3), 281–92.	Nurses play an important role in the early detection of inflammatory changes in the joints, the interpretation of laboratory test results and the continuation of pharmacological treatment. Nurse-delivered care can reduce hospital admissions and lower the cost of care.
Primdahl J., et al. [47]	The Impact on Self-Efficacy of Different Types of Follow-Up Care and Disease Status in Patients with Rheumatoid Arthritis–A Randomized Trial	*Patient Education and Counseling* 2012, 88(1), 121–128.	Identification of psychosocial problems of patients with rheumatological diseases by nurses and their support are important elements of nursing care for patients and their families. Nurse-led interventions can improve patients’ self-efficacy to cope with the disease in their daily lives.

EULAR recommendations show that APNs are able to independently provide effective follow-up of patients with chronic rheumatoid disease, with control of the risk factors for relapse and the early detection of complications. According to patients, care provided by nurses can improve their quality of life and enhance satisfaction with medical services through longer and more detailed consultations [40]. From the perspective of a healthcare system, nurse-delivered care can reduce hospital admissions, and lower the cost of care and support team coordination [45]. Patients value the competence of nurses, especially APNs, and patient trust in the family physician is the most important factor influencing the perception of the importance of care provided by nurses. Patient perspectives provide important insights into the implementation of APNs into the healthcare system. It enhances nurses’ sense of self-efficacy, resulting in improved nursing services and higher quality of care [46,47].

In conclusion, it is important to bear in mind that all medical services are defined by particular healthcare systems. Regardless of the formal conditions, a precise definition of the roles and tasks of particular professional groups should optimize the healthcare provided. This is intended to result in prompt diagnosis, and in effective, safe therapy with a consideration of the impact of the chronic disease on the patient’s functioning and immediate environment (Figure 2).

## 5. Limitations

This paper focused on recommendations that aim to shape the way nursing is delivered in the future. Therefore, the practical significance of the EULAR recommendations will become apparent several years after their implementation and after empirical research presenting both the subjective and objective scope of activities is carried out on the basis of the presented guidelines.

## 6. Conclusions

The results of our review and, above all, the EULAR recommendations, allow us to conclude that the education of nurses in the specialty of rheumatology is the key factor for developing advanced nursing competencies relevant to patient care. In addition, the accessibility of professional nursing care for patients and their families facilitates self-management and increases satisfaction with care in rheumatic diseases. Being a member of a multidisciplinary healthcare team in the care of patients with rheumatic conditions requires the nurse to be as competent as an advanced practice nurse. In order to develop and implement the role of the advanced practice nurse, it is necessary to establish clear formal regulations, and to implement standards that include both education and the comprehensive provision of medical services.

## Figures and Tables

**Figure 1 healthcare-11-02434-f001:**
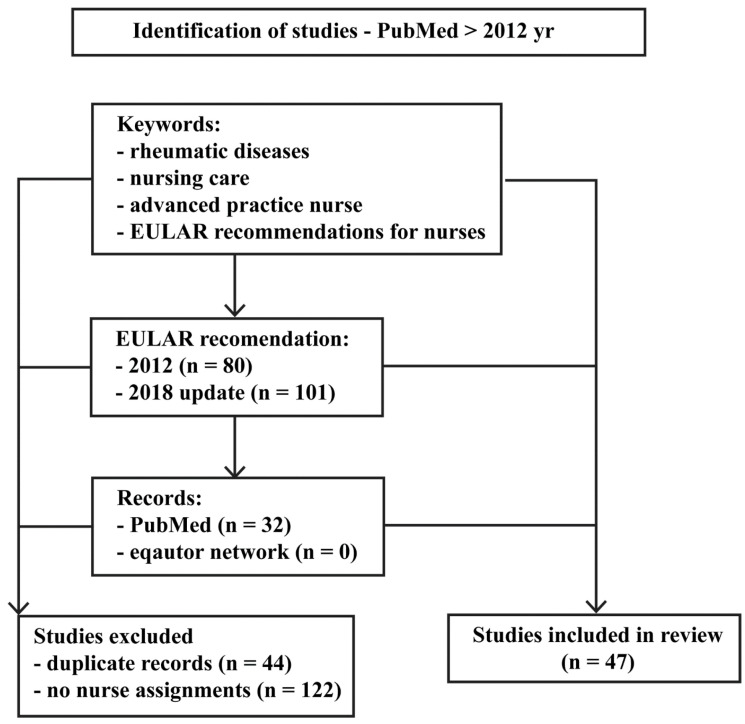
The chart for identifying articles for inclusion in qualitative analysis.

**Figure 2 healthcare-11-02434-f002:**
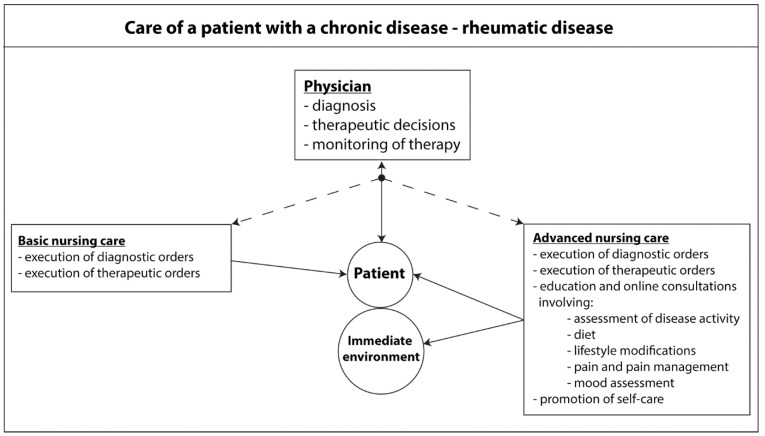
Care of a patient with a chronic rheumatic disease.

**Table 1 healthcare-11-02434-t001:** The nurse’s role in the care of patients with rheumatologic conditions, as derived from the EULAR 2018 recommendations.

Three Overarching Principles				
1. Rheumatology Nurses * Are Part of a Healthcare Team				
2. Rheumatology Nurses Provide Evidence-Based Care				
3. Rheumatology Nursing Is Based on Shared Decision-Making with the Patient				
Eight Recommendations				
Type of Recommendation	Aim of Recommendation	Level of Evidence ^1^	Grade of Recommendation ^1^	Level of Agreement ^2^ (Mean SD and Range; 0–10)
1. Patients ** should have access to a nurse	Need-based patient education to improve knowledge of the rheumatic disease and its management.	1B	A	10.0 ± 0.2 [9–10]
2. Patients should have access to nursing consultations/advice	To enhance patients’ satisfaction with care.	1A	A	9.7 ± 0.6 [8–10]
3. Patients should be able to access the nurse quickly.	To enable patients to receive tailored support (including online consultations/tele-health)	1B	B	9.7 ± 0.6 [8–10]
4. Nurses should participate in comprehensive follow-up of the patient’s disease (managing the patient’s disease activity).	To maintain the patient’s preferred quality of functioning during the illness and to ensure cost-effective care.	1A	A	9.7 ± 0.5 [8–10]
5. Nurses should provide support to patients with psychosocial problems.	To reduce symptoms of anxiety and depression in patients.	1B	A	9.6 ± 0.7 [8–10]
6. Nurses should support patients’ self-management skills.	To strengthen patients’ belief in self-efficacy.	1A	A	9.8 ± 0.4 [9–10]
7. Nurses should have access to undertake continuing education in the specialty of rheumatology.	To improve knowledge and skills.	2C	B	9.8 ± 0.7 [7–10]
8. Nurses should be encouraged to undertake extended/advanced roles following specialized training, in compliance with national regulations.	To provide comprehensive therapeutic, caring and educational services for patients.	1A	A	9.7 ± 0.6 [8–10]

* Nurses providing care to patients with rheumatic conditions. ** Patients with various rheumatic conditions. ^1^ According to the Oxford Centre for Evidence-based Medicine—CEBM ‘Levels of Evidence 1’. ^2^ Expert agreement achieved by all members of the task force upon the consensus meeting.

## Data Availability

The datasets used and/or analyzed in the current study are available from the corresponding author on request.

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
