# Peer review of "The Role of the Nurse in the Care and Management of Patients with Rheumatic Diseases Arising from the Current EULAR Recommendations: A Literature Review"

_healthcare, 2023, doi:10.3390/healthcare11172434_

Round 1

Reviewer 1 Report (Previous Reviewer 1)

I have no other comments.

Author Response

Thank you so much.

Reviewer 2 Report (Previous Reviewer 2)

No further comments.

Thank you. Good Luck :) 

Author Response

Thank you so much.

Reviewer 3 Report (Previous Reviewer 3)

Why do focus on the last ten years? It seems like a large body of literature may be lost. This seems to be done purely for convenience. 

The summary table should include more details e.g., a summary of the included articles in the review 

Minor editing of English language required

Author Response

The Authors of the article would like to thank all Reviewers for their evaluation and comments. The table below presents suggestions for supplementing the content of the article based on the suggestion of Review No. 3.

Review No. 3

Reviewer Comments

Authors' responses

Why do focus on the last ten years? It seems like a large body of literature may be lost. This seems to be done purely for convenience. 

Explanation for the Reviewer: In the methods section (lines 110-118; green) we explain in detail why we started searching for works in the scientific resource database in 2012. This paper presents the role of nurses in the care and management of a patient with rheumatic diseases, which results from the current recommendations EULAR. In 2012, this organization developed the first recommendations for nurses regarding their participation in the treatment and care of patients with rheumatic diseases, which were updated in 2018. These recommendations were the result of a review of scientific data, but above all the consensus of rheumatology experts, including nurses and patients. The aim of our work is to promote EULAR recommendations for rheumatology nurses. Therefore, we decided that our review of data on the role of a nurse in the treatment and care of a rheumatological patient should also start from the time of the first EULAR recommendations, i.e. from 2012.

The summary table should include more details e.g., a summary of the included articles in the review 

In line 216 of the text, there is table 2 (yellow) summarizing the results of the literature review in the context of the discussed recommendations.

Table 2 is also included below the reply letter.

Comments on the Quality of English Language

Minor editing of English language required

The text of the thesis was checked by a person competent in the field of English.

Table 2. Summary of selected items of the literature review in terms of overriding principles and detailed EULAR recommendations regarding the care of rheumatological patients by nurses

Author

Title

Journal, year 

Summary of research results

Perkin K.

Nurse practitioners and interprofessional collaboration.

Journal of Interprofessional Care 2011, 25(4), 243–4.

In the opinion of the patient, care provided by nurses is effective, safe and easily accessible. Rheumatology nurses are important members of the medical team caring for the patient.

Riley L., et al.

The role of nurse practitioners in delivering rheumatology care and services: Results of a U.S. survey.

Journal of the American Association of Nurse Practitioners 2017, 29(11),673–681.

Rheumatology nurses work in an interdisciplinary medical team, they constantly cooperate with the patient and his family in the field of care and treatment.

van Eijk-Hustings Y,. et al.

Dissemination and evaluation of the European League against rheumatism recommendations for the role of the nurse in the management of chronic inflammatory arthritis: results of a multinational survey among nurses, rheumatologists and patients.

Rheumatology 2014, 53 (8), 1491–6.

In professional practice, nurses use various sources of knowledge, i.e. scientific evidence, including protocols and guidelines, as well as their own experience gained while working with patients and their families.

Fusama M., et al.

Survey on attitudes regarding EULAR recommendations for the role of nurses involved in medical care of patients with chronic inflammatory arthritis in Japan.

Modern Rheumatology 2017, 27(5), 886–93.

In patient care, nurses rely on a variety of sources of knowledge, including scientific evidence, protocols, and guidelines. They are also guided by their own nursing experience resulting from the territorial context and take into account the needs and aspirations of patients.

Barbosa L., et al.

Applicability of the EULAR recommendations on the role of the nurse in the management of chronic inflammatory arthritis in Portugal.

Acta Reumatologica Portuguesa 2013, 38,(3), 186–91.

Solomon D.H., et al.

Roles of nurse practitioners and physician assistants in rheumatology practices in the US.

Arthritis Care & Research 2014, 66(7), 1108–13.

The care provided by rheumatology nurses is effective and takes into account the health needs, values and preferences of patients. Patient education by nurses covers areas related to the disease, such as pain, quality of life, and adherence.

Walker J.

Rheumatoid arthritis: role of the nurse and multidisciplinary team.

British Journal of Nursing 2012, 21(6), 334, 336–9.

Patients are satisfied with the nursing care. They have a positive experience of maintaining a competent relationship with the nurse.

Morgan S. & Yoder L.H.

A concept analysis of person-centered care.

Journal of Holistic Nursing 2012, 30(1), 6–15.

The holistic and professional approach of nurses to care inspires the trust of rheumatology patients in the knowledge of nurses, and also strengthens the supportive style of mutual relations.

Larsson I.

Nurse-led Care and Patients as Partners Are Essential Aspects of the Future of Rheumatology Care.

The Journal of Rheumatology 2017, 44(6), 720–722.

In the patients' opinion, teleconsultation by nurses provides them with a sense of individual support from a competent healthcare team. The quality of service provided by nurses is comparable to being observed by a physician in terms of disease control.

Frølund J.C. & Primdahl J.

Patients’ experiences of nurse-led screening for cardiovascular risk in rheumatoid arthritis

Musculoskeletal Care 2015, 13(4), 236–47.

Rheumatology nurses have an important role in the early detection of inflammatory changes in the joints, the interpretation of laboratory test results, the follow-up of pharmacological treatment, and the monitoring of patients' adherence to vaccination schedules.

Grønning K., et al.

Patients’ confidence in coping with arthritis after nurse-led education; a qualitative study.

BMC Nursing 2016, 15(28), 1–8.

Patients appreciate the opportunity to discuss various issues affecting their mental state with the nurse. Nurses provide support to patients with psychosocial problems in order to reduce the symptoms of anxiety and depression and motivate them to function effectively in the disease.

Robinson S., et al.

A national survey of nurse training: confidence and competence in educating patients commencing methotrexate therapy.

Musculoskeletal Care 2017, 15(3), 281–92.

Nurses play an important role in the early detection of inflammatory changes in the joints, the interpretation of laboratory test results and the continuation of pharmacological treatment. Nurse-delivered care can reduce hospital admissions and lower the cost of care.

Primdahl J., et al.

The impact on self-efficacy of different types of follow-up care and disease status in patients with rheumatoid arthritis–a randomized trial.

Patient Education and Counseling 2012, 88(1), 121–128.

Identification of psychosocial problems of patients with rheumatological diseases by nurses and their support are important elements of nursing care for patients and their families. Nurse-led interventions can improve patients' self-efficacy to cope with the disease in their daily lives.

This manuscript is a resubmission of an earlier submission. The following is a list of the peer review reports and author responses from that submission.

Round 1

Reviewer 1 Report

The topic is of particular interest since indeed, nurses play an increasingly important role to the complex management and monitoring of patients with rheumatic diseases. My enthusiasm is somewhat lessened by the difficulty to understand exactly the scope of this article unless the Authors articulate it more clearly. My understanding is that first, they attempted to perform a systematic literature review which however, yielded no relevant hits. Therefore, they went to to decompile the EULAR recommendations. This is not very clear and also, the EULAR (updated 2018) recommendations should be sufficiently cited throughout the text, Table etc. The second major point is that cited literature tends to focus on inflammatory arthritides such as rheumatoid. How about other autoimmune rheumatic diseases including connective tissue diseases? 

Author Response

Authors’ Responses (highlighted in yellow)

The authors thank the Reviewer for appreciating the role of nurses in the treatment and monitoring of patients with rheumatoid diseases. Responding to the reviewer's suggestions, we suggest adding the following text to the content of the article before the purpose of the work:

“Management of patients with rheumatic diseases has changed dramatically over the last decade and involves early initiation of intensive treatment and close monitoring of disease activity until remission is achieved. Similarly, in less common connective tissue diseases, which are characterized by immune dysfunction, vascular damage and changes in many internal organs, treatment requires a comprehensive approach to patients.

Due to the large spectrum of clinical symptoms that affect the quality of life of patients with rheumatic diseases, including connective tissue, the role of professional nursing care is essential. It provides an individual approach to the patient and gives him a sense of security and confidence in the disease. In view of the above data, the improvement of services and areas of competence for rheumatology nursing becomes necessary.”

11. Riley, L.; Harris, C.; McKay, M.; Gondran, S.E.; DeCola, P.; Soonasra, A. The role of nurse practitioners in delivering rheumatology care and services: Results of a U.S. survey. Journal of the American Association of Nurse Practitioners 2017, 29(11),673–681.

Reviewer 2 Report

Dear Authors,

Thank you for presenting an interesting manuscript with a well justified research question/topic. I do have some feedback and questions I would like answered before I can recommend this paper for publication.

My main concern at this stage is that the study design is a systematic review but is missing some fundamental reporting and information that is typically expected to be presented in a SR review. I also have a few issues with the referencing style that I think needs attention.

Please see below the following comments:

Line 36: please rephrase this to “Rheumatic diseases are also a significant cause of chronic illnesses in the paediatric population”.

Line 40: I would recommend referencing this statement with relevant data to boost the validity of what you are saying.

Line 62: Advanced Practice Nurse should be abbreviated.

Line 87: Is there a reason the search strategy started in 2012? If so please justify. Is this to do with lines 140-142? If so I would mention this earlier in the methods.

Lines 94 to 96: Text is a different size. Please fix.

Methods: Any results of your search strategy need to be moved to the results section of your manuscript not the methods.

Also typically in a systematic review you will specifically outline the inclusion and exclusion criteria. This will need to be added to your methods. I noticed this is said in lines 122-124 but what are your exclusions?

Lines 104-105: Again why was it started in 2012? Please justify.

Line 125: were any papers excluded for not being written in English?

Lines 183-187: This is a small paragraph. I wonder if it could be joint by the proceeding one?

Lines 204-210: These lines need to be referenced as they refer to research that other authors have completed.

Line 209: How do nurses play a role in the early detection of inflammatory changes in joints? May you please be more specific on how this achieved. This information would also need to be referenced.

Line 218: Please reference.

Lines 222-225 need to be referenced.

The referencing style in this manuscript is interesting. Authors have seemed to provide all referencing at the end of the paragraph rather than citing each piece of information with the corresponding reference. Or at least it seems this is the case. If this is the case you will need to adjust this throughout the manuscript. From what I can see it is mainly in the discussion that this error has occurred. So overall, it is difficult to know what pieces of information are actually referenced or not.

Limitations Section: Saying both PRISMA and AGREE failed to produce results is misleading and a bit confusing. Do you mean that your search strategy was unable to detect any papers that qualified for inclusion? I also noticed in the results that you stated a PRISMA could not be included as no studies were included. I don’t think this is necessarily true? I think you should include a PRISMA diagram to should how many studies you retrieved, how many were duplicates, how many were included for full text review/screening and then there reasons for rejection. All of this is not clear from what I can see and would need to be included.

Line 285: Also sorry what does a negative search result mean? I’m a bit confused because the next line states that you have included studies?

Another point is that the paper is a systematic review but doesn’t explicitly state this in the title. Is there a reason for this?

Conclusion: Remember that the conclusion should summarise the findings in your systematic review first then possibly end with a recommendation or two. If indeed there is a lack of evidence as shown by your systematic review search then this should be the forefront of your conclusion.

Author Response

The authors of the article would like to thank all Reviewers for their valuable comments. The table below shows changes to the content of the article based on the suggestions provided.

Reviewer 3 Report

The manuscript is a review article, and the authors have done a good job of synthesizing all the versatile findings from different articles and providing a supportive discussion and conclusion. However, I have some concerns as outlined below:

The Introduction needs to be elaborated more to highlight the significant contribution of this review

the objectives need to be revised too to highlight the purpose of this review to SYSTEMATICALLY review the relevant literature 

Why focusing on the last 10 years? It seems like a large body of literature may be lost. This seems to be done purely for convenience, and I doubt it will result in a better or more convincing piece of work.

As written in its current form, it does not seem a systematic review but a narrative review. The systematic review should follow the PRISMA checklist

The authors report that this systematic review was developed using PRISMA.  It does not appear to have been registered with PROSPERO.  There is no research question developed with a PICO or other common methods for systematic reviewand no clear search strategy provided.

The search strategy is not formal and is thus not replicable. I would doubt the authors could get to the same papers if they followed the description of their paper. This is an example of an appropriate search strategy: https://guides.lib.umich.edu/c.php?g=283340&p=2126706

QUALITY ASSESSMENTAre the outcomes of this included in a table (if so, mention the table here)?

ELIGIBILITY CRITERIA Which articles were included – was only original research, or did the authors include secondary research?  

"The researcher" or "the researchers" is used throughout for action regarding identifying the articles. Specifically, who did what? Also, there is no clear information on how many reviewers were. Normally, there are at least two, but if many papers are returned they both evaluate a sample and re-assess the strategy if there is not high agreement.

This is more of a personal dislike, but I think it is worth mentioning. I am always unconvinced by papers of this type (a systematic review without a meta-analysis). If the question is focused, a meta-analysis should be at the end of it. Otherwise, one ends up with loads of information that is very difficult to summarise and reach a conclusion from.

DISCUSSION. Some references are needed to support the discussion or even a reference to the summary table.

Author Response

(The authors gave the same response as above.)

Round 2

Reviewer 2 Report

Thank you for addressing the comments I have had so far. Please see the follow-up to the adjustments made so far:

So there remains some fundamental problems with the reporting of this "systematic review". First the title doesn't explicitly report that this study is a systematic review which I think is wrong. The authors have states the study design is a systematic review, however, the way it is presented is resembles more of a literature review not systematic.

I still don't understand what is meant by PRISMA and AGREE failed to produce results? You need to be explicit to the audience knows what you are talking about. If your search resulted in no included studies then you should report this in the results. I also don't particularly like the term "negative search result" I would remove this statement or explain what this means. 

In this paper you have referenced PRISMA but have not provided a PRISMA flow diagram. I understand that your search has not provided any included studies but that doesn't mean you don't provide a flow diagram. If you are not willing to provide a diagram showing the included studies/duplicates then excluded studies and reasons for those exclusions then I suggest you not reference PRISMA and remove systematic review as your study design.

Moreover, it is not clear what authors screened titles and abstracts and if there were any disagreements. Was there any disagreements or discrepancies in decisions being made when screening articles? If so this needs to be reported. It is very odd that this has not been reported. 

I have noticed another issue. According to your search it was conducted in Jan 2021 which means it is currently over 2 years old. Typically, searches should NOT be more than 12 months old when trying to publish results. Authors will need to rerun the search from Jan 2021 to current to make sure further studies aren't left out. Secondly, for a systematic review your search strategy does is not robust. I would strongly suggest authors to review this strategy and possibly have a research librarian confirm if this strategy is appropriate.

Overall, the authors need to consider whether this paper should be listed as a systematic review or a literature review. At this stage the current state of the paper does not qualify as a systematic review. Therefore, I cannot recommend this paper for publication. However, if the authors are willing to make major changes to that I have mentioned above then I'm happy to review it again. Alternatively you may re-state this paper as a literature review instead and update the search again for me to accept for publication.

Author Response

Dear Reviewer,

We kindly thank for all valuable suggestions.

Reviewer 3 Report

I am unimpressed by the revisions made.

Most of my carefully considered inputs have been ignored

Author Response

We kindly thank the Reviewer for taking the time to review our work and for his valuable suggestions in the subsequent stages of the review. 

Round 3

Reviewer 2 Report

Dear Authors,

Thank you for being patient during this peer review process. I am happy to recommend your manuscript for publication based on the following minor changes:

- Lines 95-96. I would prefer if you stated: "The chosen review style was literature review given the uncertain nature of any eligible studies"

- Lines 133, 142, 202 still have "systematic review". Please change this.

- The last comment I have pertains to when your search was conducted last. Lines 110-117. The start of this section says that you search the databases in January 2021. And so your search encompasses between Jan 2012 and Dec 2020. What I'm trying to say is if you last searched in Jan 2021 how do we know if there are no eligible articles between Feb 2021 to May 2023? If you only searched the databases once and the last time was Jan 2021 it means that your search is out of date. I'm sorry if are not willing to update your search then I can't recommend for publication. If you have updated your search recently please provide how many additional papers you have screened and when the search was done.

Author Response

Once again, we would like to thank the Reviewer for the time he finds for our article, for valuable suggestions and favor for our answers. 
